# Excess Nitrogen in Temperate Forest Ecosystems Decreases Herbaceous Layer Diversity and Shifts Control from Soil to Canopy Structure

**Frank S. Gilliam**

Department of Biology, University of West Florida, Pensacola, FL 32514, USA; fgilliam@uwf.edu;
Tel.: +1-850-474-2750

**Abstract:** Research Highlights: Excess N from atmospheric deposition has been shown to decrease plant biodiversity of impacted forests, especially in its effects on herbaceous layer communities. This work demonstrates that one of the mechanisms of such response is in N-mediated changes in the response of herb communities to soil resources and light availability. Background and Objectives: Numerous studies in a variety of forest types have shown that excess N can cause loss of biodiversity of herb layer communities, which are typically responsive to spatial patterns of soil resource and light availability. The objectives of this study were to examine (1) gradients of temporal change in herb composition over a quarter century, and (2) spatial patterns of herb cover and diversity and how they are influenced by soil resources and canopy structure. Materials and Methods: This study used two watersheds (WS) at the Fernow Experimental Forest, West Virginia, USA: WS4 as an untreated reference and WS3 as treatment, receiving 35 kg N/ha/yr via aerial application. Herb cover and composition was measured in seven permanent plots/WS from 1991 to 2014. In 2011, soil moisture and several metrics of soil N availability were measured in each plot, along with measurement of several canopy structural variables. Backwards stepwise regression was used to determine relationships between herb cover/diversity and soil/canopy measurements. Results: Herb diversity and composition varied only slightly over time on reference WS4, in contrast to substantial change on N-treated WS3. Herb layer diversity appeared to respond to neither soil nor canopy variables on either watershed. Herb cover varied spatially with soil resources on WS4, whereas cover varied spatially with canopy structure on WS3. Conclusions: Results support work in many forest types that excess N can decrease plant diversity in impacted stands. Much of this response is likely related to N-mediated changes in the response of the herb layer to soil N and light availability.

**Keywords:** herbaceous layer; excess nitrogen; canopy structure; temperate forests

## 1. Introduction

The herbaceous layer is increasingly acknowledged for its significant contribution to the integrity of the structure and function of forest ecosystems [1–3]. This has likely contributed, in part, to the notable increase, in recent decades, in research activity of plant ecologists investigating the dynamics of forest herb layer communities. Based on the number of publications, the new millennium has witnessed an unprecedented increase in research efforts. The number of papers published on the herb layer in the current, incomplete decade alone exceeds the total number of papers published in the 20th century by nearly 100% [2].

Along with growing awareness of the essential role that the herb layer plays in forest ecosystems is an increase in studies investigating the ecological factors that influence its dynamics, including spatial and temporal variation in species composition and aboveground cover/biomass. A review of this

expanding literature reveals sharp contrasts among the forest types studied, including widely varying land use history, that preclude broad generalizations regarding specific ecological drivers that help shape forest herb communities. Certainly, any list of abiotic factors affecting these communities would include light and moisture, especially as mitigated by the overstory canopy, as well as temperature regimes [4]. Temperature responses are of particular interest with their implications for the future of forests in the context of climate change [5,6].

Although the utility of considering stratification of vegetation in forest communities has been questioned (see Parker and Brown [7] and discussion therein), it is clear that solar radiation attenuates through a forest canopy in ways that that alter both the quantity and quality of light reaching the forest floor. Light availability is the most spatially and temporally variable component of the environment of the forest floor, varying at numerous levels over time and space, in a manner described as a *dynamic mosaic* [8], penetrating the canopy and reaching the forest floor in sunflecks, i.e., mosaics of discrete patches of light varying in size and distribution over time scales from the diurnal to the seasonal. In addition, soil resources, including nutrients and moisture, are often spatially quite heterogeneous in ways that influence the dynamics of the herb layer [9–11].

Given the increased interest in the importance of the herb layer to maintaining the structure and function of forest ecosystems, it is not surprising that considerable research has been devoted to understanding what environmental factors most sensitively affect herb layer composition and cover. Some of the more extensive earlier work was carried out by Rogers [12–16] in the 1980s in conifer and hardwood forests of the northern U.S. and Southern Canada. This work revealed that there can be great interannual variability in herb cover. He also concluded that soil fertility was more important than climate variables in influencing forest herb communities. In one of the more complete studies of vegetation recovery following disturbance, however, Reiners [17] showed that community reorganization was largely driven by light availability, as *Rubus* spp. increased and then sharply decreased during canopy redevelopment. Using a database of studies investigating effects of resources on herb layer diversity, Bartels and Chen [9] concluded neither resource availability nor heterogeneity solely influences herb layer dynamics. Reich et al. [18] found that spatial heterogeneity in the light environment most directly affect herb richness in southern boreal forests. This was confirmed for both richness and herb cover by Kumar et al. [19] studying similar stand types, with the same response found for biomass [20].

In addition to these components of the ambient environment of the forest floor are anthropogenic influences, virtually all of which represent threats to forest health, particularly regarding the herb layer of impacted forests, including the effects of excess N from atmospheric deposition [21–23]. Gilliam et al. [24] examined a quarter century of experimental additions of N on an entire watershed (WS) at the Fernow Experimental Forest (FEF), West Virginia, to study the effects of excess N on temporal and spatial dynamics of the herb layer of a Central Appalachian hardwood forest. This study involved nearly annual monitoring of herb layer composition and cover of permanent plots from 1991 to 2014 on both an N-treated watershed and the long-term reference watershed at FEF (WS4), a ~100 year old mixed hardwood forest. They found a pronounced shift in herb layer composition over the 25 years of experimental N additions to the treatment watershed, and determined that such change arose from increases in a nitrophilic species (i.e., *Rubus allegheniensis* Porter that competitively excluded numerous N-efficient herbaceous species, ultimately decreasing plant diversity.

This response supported the N homogeneity hypothesis, which predicts that excess N deposition to forest ecosystems increases the spatial homogeneity of N by decreasing natural patchiness of N availability essentially by filling in the low-N matrix within which discrete high-N patches occur [25]. As a result, temporal increases in atmospheric inputs of N should increase N availability within this matrix to approach that within the patches of high fertility. Nitrophilic plant species of the forest herbaceous layer then increase in dominance, outcompeting the more numerous N-efficient species and decreasing biodiversity of the forest, up to 90% of which is represented by the herb layer [1].

To further investigate this response, Walter et al. [26] carried out field studies on the response of *R. alleghentensis* to variation in light and N availability. They compared relative cover of *R. alleghentensis* in N-treated WS3 and another untreated watershed at FEF (WS7) and among N-fertilized and unfertilized experimental plots; both approaches utilized canopy openness as a covariate. The ex situ experiment used a two-way factorial design, measuring leaf area with two levels of N and three of light. Results of both approaches were consistent in revealing that the effects of N availability on cover were significantly mitigated by availability of light.

The objectives of this paper are two-fold. The first is to further analyze data from the Gilliam et al. [24] study to more specifically focus on gradients of change in herb layer composition and temporal variation in indices of biodiversity in the context of forest response to excess N. The second includes previously unpublished canopy structural data to examine spatial patterns of herb cover and diversity and how they are influenced by several metrics of canopy structure (many of which are indicative of light availability to the forest floor [27]) and soil resources in both watersheds, with a particular focus on how excess N might alter these relationships.

## 2. Methods

### 2.1. Study Site

This study was carried out at the Fernow Experimental Forest (FEF), Tucker County, West Virginia (39°03′15″ N, 79°49′15″ W), as part of a long-term study on the effects of chronic additions of N on the structure and function of central Appalachian hardwood forest ecosystems. Fernow Experimental Forest is a ~1900 ha area of the Allegheny Mountain section the unglaciated Allegheny Plateau. Precipitation for FEF averages ~1430 mm yr$^{-1}$, with precipitation generally increasing through the growing season and with higher elevations [24].

Two watersheds were used for the location of sample plots: WS3 and WS4, with WS3 serving as the treatment watershed, receiving aerial additions of $(NH_4)_2SO_4$, and WS4 serving as reference watershed. Applications of $(NH_4)_2SO_4$ to WS3 began in 1989 and are currently on-going. Aerial applications of $(NH_4)_2SO_4$ are made three times per year, and historically have been administered by either helicopter or fixed-wing aircraft. March and November applications are 33.6 kg/ha of fertilizer, or 7.1 kg/ha of N. July applications are 100.8 kg/ha fertilizer (21.2 kg/ha N). Stands on WS3 were ~45 yr-old at the time of most recent sampling in this study (2014); these are even-aged and developed following clearcutting. WS4 currently supports even-aged stands >100 yr old.

Study watersheds support mixed hardwood stands. Overstory dominant species include sugar maple (*Acer saccharum* Marsh.), sweet birch (*Betula lenta* L.), American beech (*Fagus grandifolia* Ehrh.), yellow poplar (*Liriodendron tulipifera* L.), black cherry (*Prunus serotina* Ehrh.), and northern red oak (*Quercus rubra* L.). In 1991, species composition of the herbaceous layer was quite similar between watersheds, despite differences in stand age, including species of *Viola*, *Rubus*, mixed ferns, and seedlings of striped maple *Acer pensylvanicum* L. and red maple *A. rubrum* L.

### 2.2. Field Methods

The herbaceous layer was sampled in five circular 1 m$^2$ sub-plots within each of seven circular 0.04 ha permanent sample plots using methods described in Gilliam et al. [24]. Briefly, all vascular plants ≤1 m in height in each subplot were identified to species (sometimes to genus) and visually estimated for cover (%); see Walter et al. [28] for detailed description of this method. This was carried out in the first week of July each of the sample years 1991, 1992, 1994, 2003, and annually from 2009 to 2014, for a total of 10 sample years over a 24-year period representing 26 years of N treatment on WS3. When used for regression analyses, soil data were taken from Gilliam et al. [29], including soil moisture (%), extractable $NH_4^+$ and $NO_3^-$, and net N mineralization and nitrification. Soil moisture was determined gravimetrically, $NH_4^+$ and $NO_3^-$ were measured colorimetrically following 1N KCl extraction, and net N mineralization and nitrification were determined using in situ

incubations. These were for mineral soil only (O horizon excluded) that was taken to a 5-cm depth, with measurements made on a monthly basis. For this analysis, data from July 2011 were used to align with the herb layer and forest canopy measurements.

Forest canopy measurements were made in July 2011 with a Riegl LD90-3100VHS-FLP laser rangefinder (operating in first-return mode at 890 nm and 2 kHz, laser safety class I) mounted to the front of a frame at 1 m above the ground and manually pointed upward, making 2000 measurements per second. Data were transferred through a serial cable to laptop [30]. Using constant walking speed, locations of each range measurement were estimated from its sequence in the data file. Generally, distances between measurements were <1 cm, with the spot size of the laser beam being 4–6 cm at the ranges measured.

The data files were edited to identify values that were out-of-range (e.g., when penetrating canopy openings to the sky) and remove spurious values. The edited files were processed through a program customized for grouping ranges horizontally, calculating vertical profiles (using methodology of MacArthur and Horn (1969) [31]), estimating surface area density using the overlap transformation, and assigning coordinates to each estimate. Bins used were 1 m in the horizontal and 1 m in the vertical. Resulting estimates refer to cube-shaped voxels of $1 \times 1 \times 1$ m in the x, y, and z dimensions, respectively.

### 2.3. Data Analysis

Herb layer data for the entire study period were subjected to detrended correspondence analysis (DCA), using CANOCO 4.5, for each watershed separately to assess temporal change in herb community composition. For graphical purposes, two dimensional means (from individual plot axis 1 and axis 2 values) were calculated for each year as centroids. Part of the output of DCA on CANOCO 4.5 (Microcomputer Power, Ithaca, NY, USA) are several metrics of biodiversity, including species richness and evenness and Hill and Shannon diversity indices. Changes in metrics of biodiversity of the herb layer over time, including species richness, evenness, and diversity, were assessed via Pearson product–moment correlation [32].

For the purpose of this study, the following canopy structural variables were determined: canopy area index (CAI), local outer canopy height (LOCH), rugosity, and gap fraction. Canopy area index is the sum of surface area density across all levels in a column. Local outer canopy height is the maximum surface height in a column—across all columns together these define the outer canopy surface. Rugosity is a measure of the "roughness" of the forest canopy and is the standard deviation of the mean outer canopy height. The gap fraction is the fraction of horizontal locations without any canopy surface area directly above (one minus the "cover") [30].

Potential effects of both canopy structure and soil variables on herb cover and biodiversity metrics were assessed on data from 2011 (the only year for which canopy measurements were made) using backwards stepwise regression. This procedure eliminates variables from the proposed model sequentially until all remaining variables produce F statistics that are significant at a given level of probability, in this case $p < 0.05$ [32]. This was used to identify which (if any) canopy and/or soil variables best explain spatial variation in herb diversity and cover with the following initial model:

$$Y = moist + NH_4^+ + NO_3^- + Nmin + nit + CAI + LOCH + gapfrac + rug$$

where, moist is soil moisture (%), $NH_4^+$ is extractable soil $NH_4^+$ (µg N/g soil), $NO_3^-$ is extractable soil $NO_3^-$ (µg N/g soil), Nmin is net N mineralization (µg N/g soil/d), nit is net nitrification (µg N/g soil/d), CAI is canopy area index, LOCH is local outer canopy height (m), gapfrac is gap fraction, rug is rugosity, and Y is the dependent variable. Separate runs were made for the following dependent variables: herb cover, richness, evenness, Hill diversity, and Shannon diversity.

For spatial analyses, herb layer cover, CAI, and LOCH were kriged separately in each watershed and each year using an ordinary kriging method with a spherical variogram model and global search radius in R package gstat [33]. Each model was fit using a common initial range and sill value and

interpolated onto a grid with a cell resolution of 5 × 5 m. Grids were mapped in ArcGIS using 20 equal intervals that spanned the range of kriged values for cover, CAI, and LOCH, separately, for 2011.

Temporal change in metrics of biodiversity for the study watersheds was assessed via linear correlation of individual metrics versus year, from 1991 to 2014. In addition, because of its potential as a mechanism driving change in these metrics, as just discussed, cover (%) of *R. allegheniensis* was also assessed in this manner.

## 3. Results and Discussion

### 3.1. Temporal Variation in Composition of the Herbaceous Layer

For clarity, the results of DCA are presented in two figures for each of reference WS4 and treatment WS3, the first with annual centroids labeled by year and arrows depicting temporal trends as trajectories of change in ordination space. The second figure for each watershed displays the location of prominent herb layer species for each watershed, along with unlabeled annual centroids for purposes of comparison.

As WS4 is typical of a mature second-growth stand of the central Appalachian region [34], results suggest that, although there is notable inter-annual variability, changes in herb community composition is not unidirectional over time in such stands. For example, although substantial variation occurred in the nine years between 1994 and 2003, far more occurred in the following six years to 2009; indeed, this represented a return toward increased similarity with 1991, despite the 18 year duration of this period (Figure 1). Species variation along this time trend appeared to occur across two gradients. The first of these suggests a fertility gradient, from *Vaccinium* spp. (VACC), which are well-adapted to weathered, infertile soil [35], to Smilax rotundifolia (SMRO), and then *Rubus allegheniensis* (RUAL) a nitrophilic species [26,36]. The second is likely a moisture gradient, from seedlings of overstory species (e.g., *Quercus rubra*, *Acer saccharum*, *A. rubrum*) (QURU, ACSA, ACRU) to *Laportea canadensis* (LACA) and *Dryopteris intermedia* (DRIN), which are characteristic of moist forest soils [37,38] (Figure 2).

Differences between results of DCA on temporal trends of herb composition WS4 and those of WS3 suggest the profound effects of experimental additions of N on herb layer dynamics. Although there was minimal variation along Axis 1 during the initial period of the study (1991–1994), there was a substantial shift in composition from 1994 to 2003, one which never returned toward the initial period, as was observed for WS4 (Figure 3). Similar to WS4, species variation along this time trend on WS3 occurred across two gradients. The first suggests a disturbance gradient, from *S. rotundifolia*, often associated with canopy disturbances [39], to fern species (*Polystichum acrosticoides*, *Dennstaedia punctiloba*) (POAC, DEOU), which are better adapted to less-disturbed conditions (Figure 4).

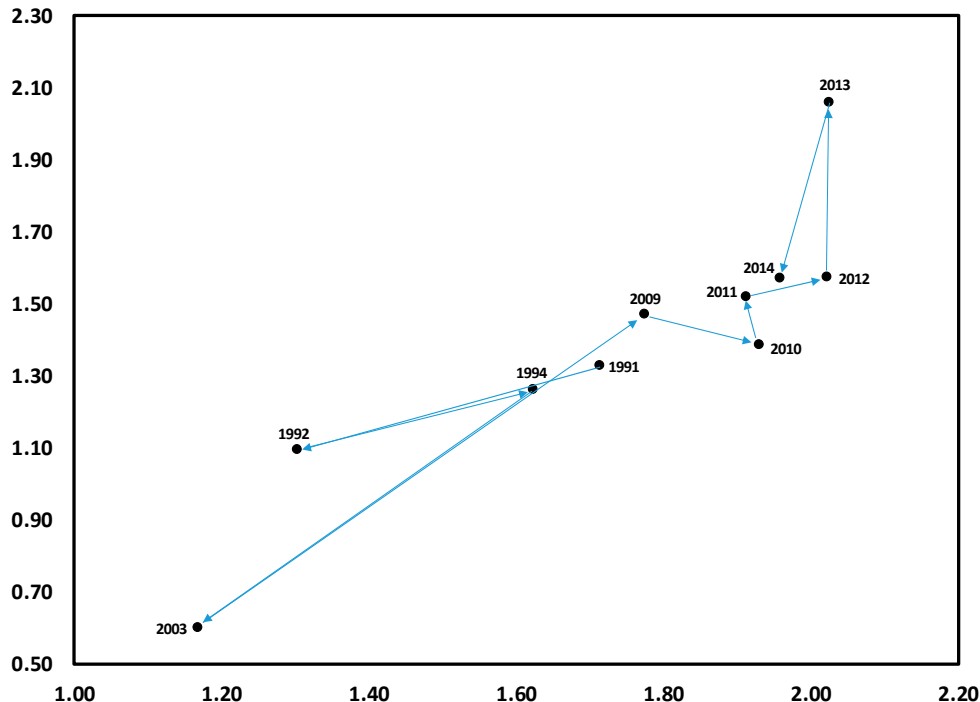

**Figure 1.** Detrended correspondence analysis (DCA) for herbaceous layer communities from 1991 to 2014 on reference watershed WS4. Data points shown along with years are centroids (two-dimensional means) for each year.

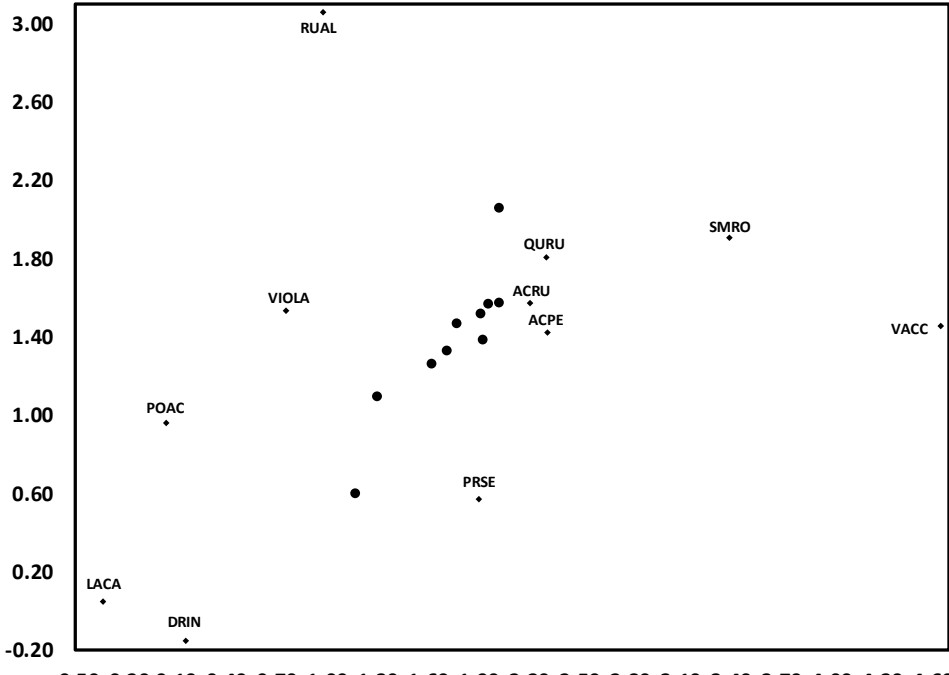

**Figure 2.** Detrended correspondence analysis (DCA) for herbaceous layer communities from 1991 to 2014 on reference WS4. Depicted are prominent species, along with centroids as shown in Figure 1. Species codes are as follows: ACPE (*Acer pensylvanicum*), ACRU (*A. rubrum*), DRIN (*Dryopteris intermedia*), LACA (*Laportea canadensis*), POAC (*Polystichum acrostichoides*), PRSE (*Prunus serotina*), QURU (*Quercus rubra*), RUAL (*Rubus allegheniensis*), SMRO (*Smilax rotundifolia*), VACC (*Vaccinum* spp.), and VIOLA (*Viola* spp.).

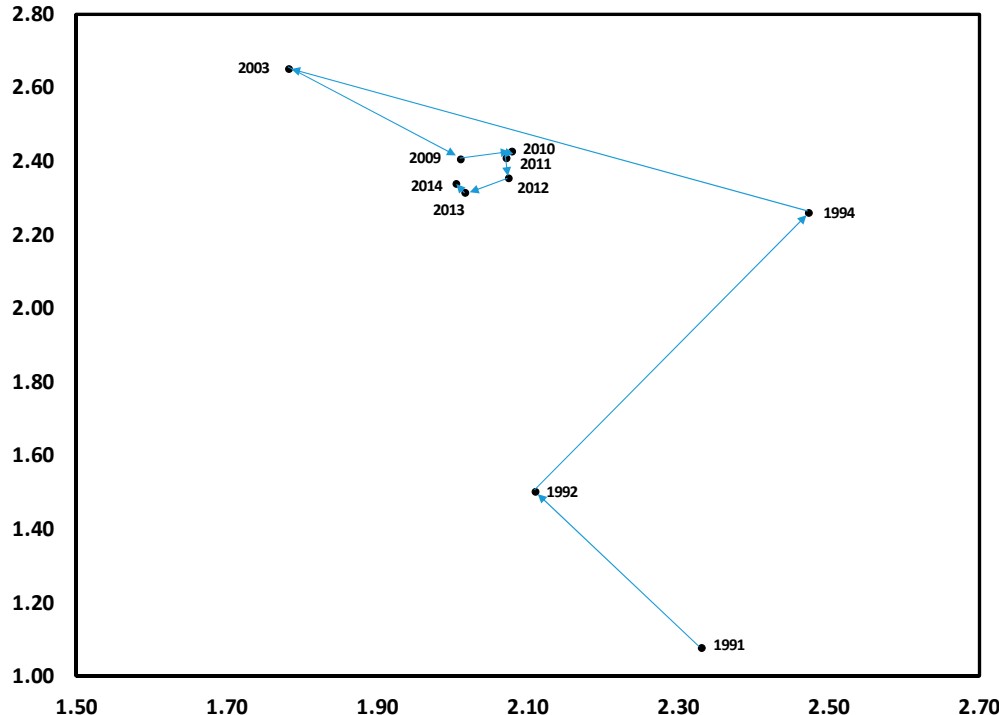

**Figure 3.** Detrended correspondence analysis (DCA) for herbaceous layer communities from 1991 to 2014 on N-treated WS3. Data points shown along with years are centroids (two-dimensional means) for each year per watershed.

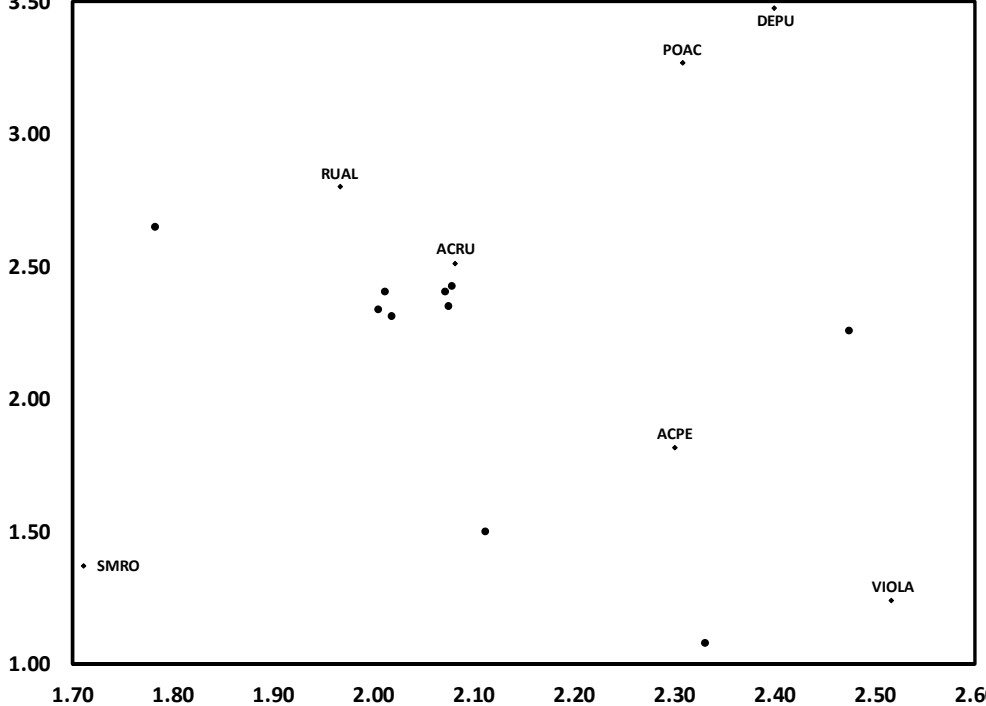

**Figure 4.** Detrended correspondence analysis (DCA) for herbaceous layer communities from 1991 to 2014 on N-treated WS3. Depicted are prominent species, along with centroids as shown in Figure 3. Species codes are as follows: ACPE (*Acer pensylvanicum*), ACRU (*A. rubrum*), DEPU (Dennstaedtia punctilobula), POAC (*Polystichum acrostichoides*), RUAL (*Rubus allegheniensis*), SMRO (*Smilax rotundifolia*), and VIOLA (*Viola* spp.).

The most notable gradient, and one that has important implications for effects of N on forest biodiversity, is that between *Viola spp.* and *R. allegheniensis*. In particular, both the direction and distance between those two species in ordination space reflects a change in dominance of the herb community from 1991 to 2014 (Figure 4). That is, the once-dominant and species-rich Viola group has been replaced by *R. allegheniensis*, supporting earlier predictions of the N Homogeneity Hypothesis: that increasing inputs of N would provide a competitive advantage to nitrophilic species, such as *R. allegheniensis*, at the expense of N-efficient species which comprise most of the biodiversity of the herb community [25]. This response is one of the more prominent mechanisms by which excess N can decrease forest biodiversity, considering that (1) there is far greater species richness among N-efficient species than among nitrophilic species, and (2) up to 90% of plant diversity of forest ecosystems is in the herb layer [40].

### 3.2. Change in Herb Diversity Metrics over Time

Species richness (S) increased significantly ($p < 0.0004$) on WS4 for the study period, suggesting that herb communities in mature second growth hardwood stands can experience the influx of new species well after 100 yr of post-disturbance succession. By contrast, species evenness (J) did not vary significantly ($p > 0.05$) with time, resulting in no change in herb layer diversity (Hill's N2). Significant increases in cover of *R. allegheniensis* did not influence any metrics of biodiversity (Table 1).

**Table 1.** Pearson product–moment correlation coefficients between species richness (S), evenness (J), and diversity (Hill's N2), and cover of *Rubus allegheniensis* (*Rubus*, in %) versus time (years) on reference WS4 and treatment WS3. Thus, significant positive and negative coefficients indicate increases and decreases, respectively, over time. Values shown are coefficients significant at $p < 0.05$. NS indicates no significant correlation.

| Metric | WS4 | WS3 |
|:---:|:---:|:---:|
| S | 0.91 | NS |
| J | NS | −0.81 |
| Hill's N2 | NS | −0.64 |
| *Rubus* | 0.86 | 0.98 |

Contrasts between watersheds, as depicted in Table 1, suggest both the profound effect of excess N on herb diversity and the principle mechanism for such an effect. As already reported for this site [24], and consistent with findings of numerous other studies [41–43], chronic additions of N to WS3 have significantly ($p < 0.05$) decreased herb layer diversity. Although both richness and evenness contribute to species diversity, these results reveal that the N-mediated loss of diversity arose from decreased species evenness, which also declined significantly ($p < 0.005$), and not species richness, which showed no change (Table 1). Combining these contrasts with the highly significant ($p < 0.0001$) increase in *R. allegheniensis* on WS3 confirm that these N-mediated declines in herb diversity are driven primarily by increasing cover of a nitrophilic species, as initially predicted by Gilliam [25] and found in several studies in contrasting forest types [21,44–46]. It is notable that *R. allegheniensis* increased significantly on WS4 over this same period, yet appeared to have no influence on herb diversity metrics. The increase is quite likely from the chronically elevated ambient levels of N deposition for this site [24], whereas the lack of effect is the much lower degree of increase on WS4.

### 3.3. Influences of Canopy and Soil Variables on Herbaceous Biodiversity and Cover

When the model was run with each of richness, evenness, and diversity separately, none was significant ($p > 0.05$), suggesting that neither canopy nor soil variables measured herein influences herb layer biodiversity for either watershed, regardless of experimental treatment. By contrast, regressions for cover revealed significant, though contrasting, model outcomes for both watersheds (Table 2). Indeed, such contrasts were strongly suggestive of effects of added N on herb layer cover. Results indicate that for reference WS4, herb cover was influenced by soil resources, rather than canopy structural variables, especially pools of available $NH_4^+$ and $NO_3^-$ (Table 2). Model outcome for treatment WS3, however, demonstrated that canopy structural, rather than soil resource, variables were primarily influencing herb cover. Thus, in the absence of added N, cover appears to be controlled primarily by soil N, whereas chronic additions of 35 kg N/ha/yr have brought about a shift in control wherein light availability, as influenced by canopy structure, more sensitively influences herb cover in these forest stands.

**Table 2.** Backward stepwise regression for study watershed at Fernow Experimental Forest, WV, beginning with the following initial model: cover = moist + $NH_4^+$ + $NO_3^-$ + $N_{min}$ + nit + CAI + LOCH + gapfrac + rug, where cover is percent herb layer cover, moist is soil moisture, $NH_4^+$ is extractable soil N, $NO_3^-$ is extractable soil N, $N_{min}$ is net N mineralization, nit is net nitrification, CAI is canopy area index, LOCH is local outer canopy height, gapfrac is gap fraction, and rug is rugosity. See Methods for explanation of canopy structural variables, stepwise regression procedure, and units for independent variables.

| Watershed | Final Model | $r^2$ |
|:---:|:---:|:---:|
| WS4 | Cover = 60.8 − 5.3$NH_4^+$ + 12.6$NO_3^-$ | 0.85 |
| WS3 | Cover = 408.5 − 58.9CAI + 3.1LOCH − 909.6gapfrac | 0.998 |

Among the canopy structural variables, canopy area index (CAI) and local outer canopy height (LOCH) appeared to exert the strongest influence on herb cover on WS3. Neither were significantly correlated with cover on WS4, but were negatively ($r = −0.88$, $p < 0.05$) and positively ($r = 0.88$, $p < 0.05$) correlated, respectively, on WS3. Actual spatial patterns of all three variables on both watersheds were determined via kriging (Figure 5). Because CAI is the sum of surface area density across all levels in a vertical column through the canopy, it should be negatively related to light availability to the forest floor. That is, higher CAI would indicate a greater leaf area intercepting more light. The negative relationship between CAI and cover on WS3 supports this (Figure 5). By contrast, LOCH likely positively related to light availability to the forest floor, being calculated as the maximum surface height in a column, rather than being an indicator of foliar surface area. This is consistent with the positive relationship between LOCH and herb cover on WS3 (Figure 5).

Contrasts between treated and reference watersheds in the present study suggest that a similar phenomenon to that reported by Walter et al. [26] for *R. allegheniensis* (see Introduction) may have occurred for the herb community as a whole. That is, 25 years of adding N to WS3 has likely shifted herb cover toward being more sensitive to spatial variability of light incident on the forest floor due to less dependence on available soil N.

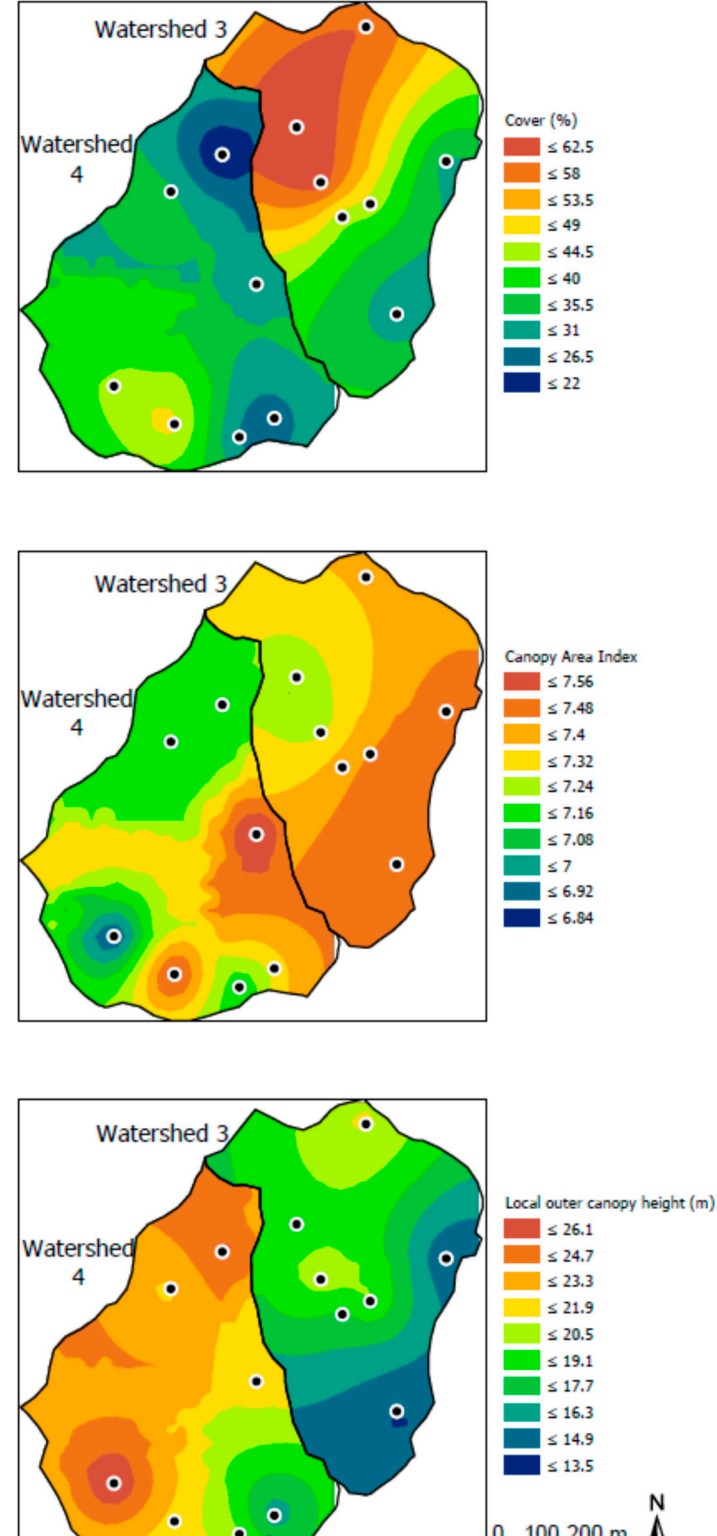

**Figure 5.** Spatial patterns of herbaceous layer cover, canopy area index, and local outer canopy height for study watersheds at Fernow Experimental Forest, West Virginia, July 2011. Shown also are locations of permanent sample plots on each watershed.

## 4. Conclusions

Differences found in this study between reference WS4 and N-treated WS3 in the nature of how the herb layer is affected by soil resources versus canopy structure highlight mechanisms of how excess N can influence forest herb communities, adding to a growing number of such studies. Results presented here further demonstrate the loss of forest biodiversity, via decreases in species diversity of herb communities, in response to simulated increases in deposition of N. In addition, this study suggests that excess N can facilitate a shift in factors controlling herb layer dynamics from variation in soil resources to variation in canopy structure.

Finally, ambient deposition of N is currently far different than that which initially created the impetus for studies of excess N on forest ecosystems. Due to the efficacy of the Clean Air Act in the United States, deposition of oxidized N has been declining in recent decades, particularly in the eastern U.S. [47,48]. In contrast, deposition of reduced N has increased over the same period [49]. Thus, future investigations on impacts of N on forest ecosystems should consider both the ways in which N-impacted forested regions recover toward pre-impact conditions and the responses of forests to changes in dominant chemical forms of N. Gilliam et al. [49] has suggested a hysteretic model for such recovery, which predicts a variable and, in some cases, considerable lag time before such changes may be detected. The question remains regarding how the herbaceous layer of N-impacted forests, and its response to soil and canopy structure, will change in a lower-N future.

**Funding:** This research was funded in part through United States Department of Agriculture (USDA) Forest Service, Fernow Experimental Forest, Timber and Watershed Laboratory, Parsons, W.V., under USDA Forest Service Cooperative Grants 23-165, 23-590, and 23-842 to Marshall University. Additional funding for this research was provided by USDA National Research Initiative Competitive Grants (Grant NRICGP #2006-35101-17097) to Marshall University and by the Long Term Research in Environmental Biology (LTREB) program at the National Science Foundation (Grant Nos. DEB0417678 and DEB-1019522) to West Virginia University.

**Acknowledgments:** I am indebted to numerous undergraduate and graduate students at Marshall University and West Virginia University for their excellent assistance in the field, and to Bill Peterjohn (West Virginia University) and Mary Beth Adams (USDA Forest Service) for field and logistical support. I am especially indebted to Jess Parker (Smithsonian Environmental Research Center) for all canopy structural measurements and to Chris Walter (University of Minnesota) for generating the kriged maps. The long-term support of the USDA Forest Service in establishing and maintaining the research watersheds is acknowledged.

**Conflicts of Interest:** The author declares no conflict of interest.

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
