# Peer review of "Excess Nitrogen in Temperate Forest Ecosystems Decreases Herbaceous Layer Diversity and Shifts Control from Soil to Canopy Structure"

_forests, doi:10.3390/f10010066_

Round 1

Reviewer 1 Report

GENERAL COMMENTS

This study provides interesting and useful information regarding the effects of excess nitrogen (N) on the herbaceous layer of forest ecosystems over 26 years of N treatment. The study expands on knowledge gained from previous work in the same watersheds. Over all, the manuscript was clear and concise, and presented and discussed interesting results contrasting how soil resources and canopy variables affected (or did not affect) herb layer communities in the reference watershed versus the N-treated watershed. There were numerous small grammatical errors throughout the manuscript that should be addressed; I did not attempt to comment on all of them. My specific comments are below.

SPECIFIC COMMENTS

Title:

-The title is a bit convoluted and doesn’t highlight the key findings. Recommendation: “Excess nitrogen in forest ecosystems decreases herbaceous layer diversity and shifts controls from soil- toward canopy-related variables”

Abstract:

-Varying spaces (one or two) after periods throughout manuscript

-Line 13: The objectives of this study were “to” examine…

-Line 15: “influenced”

-Lines 26-27: Awkward sentence

Introduction:

The introduction needs additional information and background that is contained elsewhere in the manuscript; please see later comments regarding what I suggest should be moved to the Introduction section. Also, please elaborate on how the current study expands on the previous studies.

-Line 33: Is “productivity” the correct word to use here?

-Line 34: Based on “the” number of…

-Line 41: Consider rephrasing “quick perusal through” (perhaps to “brief review of”)

-Line 41: …sharp contrasts among “the” forest types “studied”, including…

-Line 42: Remove “in ways”

-Line 44: Should “be” be replaced with “include”?

-Objectives: It seems that both objectives are based on further analyzing the data from the previous study, whereas the objectives suggest that only Part 1 is further analyzing the data from the previous study. Was any new data collected and used just for this manuscript? Please make this clear both here and in the Methods section.

Methods:

-Line 69: Fernow Experimental Forest (FEF)

-Line 79: How far apart were WS3 and WS4? Was there a possibility of cross-contamination from aerial N applications?

-Line 81: WS7?

-Line 81: What is the year of the most recent sampling?

-Line 82: Was the WS4 stand >100 years old in 1991 when the study started?

-Line 86: What year is referenced with the word “initially”? 1991?

-Lines 88-90: What is meant by “currently”? This information seems like it would be more appropriate in the Introduction or Results and Discussion section.

-Line 93: Remove “previously” and add …methods described “in Gilliam et al. (2016)” [10].

-Line 94: ≤1 m “in height”

-Line 96: 24-year period

-Lines 97-99: Please include the depth at which soil samples were taken and the year of the sampling for reader’s immediate reference

-Lines 103-104: Is “constant walking speed” a reasonable assumption?

-Lines 115-116: detrended correspondence analysis (DCA)

-Line 121: Please provide equations for species richness, evenness, and diversity or elaborate on how they were calculated

-Soil variables: C/N ratio of soil might be an important independent variable. Was it measured, and why wasn’t it included in model?

Results and discussion:

-Figures 2 & 4: Please include the key for the species codes

-Lines 172-173: …as shown in Figure 1.

-Lines 180-181: …as shown in Figure 3.

-Line 183: Viola spp. (codes?)

-Lines 186-187: N Homogeneity Hypothesis should be introduced and explained in the Introduction section

-Lines 194-197: Move to Methods section

-Line 201: …herb layer diversity (Hill’s N2)

-Line 209: …which “showed no change”

-Lines 209-212: The effect on R. alleghenienis was also highly significant on WS4 (not as high as for WS3, but still high – 0.86 vs 0.98); can you discuss why the effect on R. alleghenienis might have affected the other three herb metrics differently for WS3 than for WS4?

-Table 1: Define NS

-Lines 217-227: Move to Methods section

-Table 2, Lines 243-247: Should also be included in Methods section

-Lines 255-256: Remove “(again, not significant)”

-Line 259: Remove “but not WS4”

-Line 259: Add a summary sentence, something like: “That is, for WS3, more light to the forest floor meant greater herbaceous cover, a relationship that was not significant for WS4 (Figures 5, 6).”

-Lines 260-266: Move to Introduction section; include only a brief refresher/reminder of the information here.

-Line 269: Add something like …light incident on the forest floor “due to less dependency on soil available N”.

-Add discussion as to why WS3 herb cover is less dependent on soil available N now; how much more soil available N was there in WS3 compared to WS4?

-Add two figures similar to Figures 5 & 6 for Cover (%) vs NH4+ and NO3-

Conclusions:

-Lines 279-304: Move to Introduction section (note: remove “as confirmed in the present study” from Line 294). Replace with more brief conclusions highlighting the key take-away findings of the current study.

-Line 308: Replace “On the other hand” with “However”

Author Response

Author responses to reviewer comments and suggestions

Although I am pleased with the generally positive comments from both revisions, I am far more appreciative of the numerous suggestions for improvements regarding my manuscript.  Indeed, I feel that making virtually of these changes has improved its clarity and meaning.  Below are my specific responses given in red font.

Reviewer 1

Comments and Suggestions for Authors

GENERAL COMMENTS

This study provides interesting and useful information regarding the effects of excess nitrogen (N) on the herbaceous layer of forest ecosystems over 26 years of N treatment. The study expands on knowledge gained from previous work in the same watersheds. Over all, the manuscript was clear and concise, and presented and discussed interesting results contrasting how soil resources and canopy variables affected (or did not affect) herb layer communities in the reference watershed versus the N-treated watershed. There were numerous small grammatical errors throughout the manuscript that should be addressed; I did not attempt to comment on all of them. My specific comments are below.

Once again, these positive comments are appreciated.

SPECIFIC COMMENTS

Title:

-The title is a bit convoluted and doesn’t highlight the key findings. Recommendation: “Excess nitrogen in forest ecosystems decreases herbaceous layer diversity and shifts controls from soil- toward canopy-related variables”

Changed

Abstract:

-Varying spaces (one or two) after periods throughout manuscript

-Line 13: The objectives of this study were “to” examine… added

-Line 15: “influenced” changed

-Lines 26-27: Awkward sentence changed

Introduction:

The introduction needs additional information and background that is contained elsewhere in the manuscript; please see later comments regarding what I suggest should be moved to the Introduction section. Also, please elaborate on how the current study expands on the previous studies. Done

-Line 33: Is “productivity” the correct word to use here? No, changed

-Line 34: Based on “the” number of… added

-Line 41: Consider rephrasing “quick perusal through” (perhaps to “brief review of”) done

-Line 41: …sharp contrasts among “the” forest types “studied”, including… changed

-Line 42: Remove “in ways” done

-Line 44: Should “be” be replaced with “include”? yes

-Objectives: It seems that both objectives are based on further analyzing the data from the previous study, whereas the objectives suggest that only Part 1 is further analyzing the data from the previous study. Was any new data collected and used just for this manuscript? Please make this clear both here and in the Methods section. done

Methods:

-Line 69: Fernow Experimental Forest (FEF) changed

-Line 79: How far apart were WS3 and WS4? Was there a possibility of cross-contamination from aerial N applications? Watersheds were adjacent, but care is taken to avoid that problem.  Also, sample plots are quite distant from boundaries.

-Line 81: WS7? corrected

-Line 81: What is the year of the most recent sampling? added

-Line 82: Was the WS4 stand >100 years old in 1991 when the study started? Clarified

-Line 86: What year is referenced with the word “initially”? 1991? Clarified

-Lines 88-90: What is meant by “currently”? This information seems like it would be more appropriate in the Introduction or Results and Discussion section. This has been deleted/moved

-Line 93: Remove “previously” and add …methods described “in Gilliam et al. (2016)” [10]. Done

-Line 94: ≤1 m “in height” added

-Line 96: 24-year period added

-Lines 97-99: Please include the depth at which soil samples were taken and the year of the sampling for reader’s immediate reference done

-Lines 103-104: Is “constant walking speed” a reasonable assumption? Yes, in accord with the method.

-Lines 115-116: detrended correspondence analysis (DCA) added

-Line 121: Please provide equations for species richness, evenness, and diversity or elaborate on how they were calculated these were calculated using the usual equations

-Soil variables: C/N ratio of soil might be an important independent variable. Was it measured, and why wasn’t it included in model? C/N ratio was not routinely measured along with N measurements

Results and discussion:

-Figures 2 & 4: Please include the key for the species codes done

-Lines 172-173: …as shown in Figure 1. I generally avoid direct references to figures/tables

-Lines 180-181: …as shown in Figure 3. See above

-Line 183: Viola spp. (codes?) See above

-Lines 186-187: N Homogeneity Hypothesis should be introduced and explained in the Introduction section done

-Lines 194-197: Move to Methods section done

-Line 201: …herb layer diversity (Hill’s N2) done

-Line 209: …which “showed no change” done

-Lines 209-212: The effect on R. alleghenienis was also highly significant on WS4 (not as high as for WS3, but still high – 0.86 vs 0.98); can you discuss why the effect on R. alleghenienis might have affected the other three herb metrics differently for WS3 than for WS4? done

-Table 1: Define NS done

-Lines 217-227: Move to Methods section done

-Table 2, Lines 243-247: Should also be included in Methods section done

-Lines 255-256: Remove “(again, not significant)” done

-Line 259: Remove “but not WS4” done

-Line 259: Add a summary sentence, something like: “That is, for WS3, more light to the forest floor meant greater herbaceous cover, a relationship that was not significant for WS4 (Figures 5, 6).” done

-Lines 260-266: Move to Introduction section; include only a brief refresher/reminder of the information here. done

-Line 2 done 69: Add something like …light incident on the forest floor “due to less dependency on soil available N”.

-Add discussion as to why WS3 herb cover is less dependent on soil available N now; how much more soil available N was there in WS3 compared to WS4? done

-Add two figures similar to Figures 5 & 6 for Cover (%) vs NH4+ and NO3- These have replaced with kriged maps

Conclusions:

-Lines 279-304: Move to Introduction section (note: remove “as confirmed in the present study” from Line 294). Replace with more brief conclusions highlighting the key take-away findings of the current study. done

-Line 308: Replace “On the other hand” with “However” changed

Reviewer 2 Report

The present study examined spatial and temporal differences in herb biodiversity and cover, over 26 years, in two watersheds (or plots), one of which received fertilization multiple times a year, the other did not. The study further examined the relationships between various soil N metrics, soil moisture, various canopy metrics, and herb biodiversity and cover. From this study, the author was able to draw some interesting and meaningful conclusions; including that the addition of nitrogen lowers herb biodiversity, and herb cover was largely driven by nitrogen limitations in the control plot and light limitations in the plot receiving fertilization (as nitrogen was no longer limiting). The use of DCA in the analysis was appropriate.

The effects of additional N on herb layer biodiversity and cover was previously reported in the manuscript “Twenty‐five‐year response of the herbaceous layer of a temperate hardwood forest to elevated nitrogen deposition”, on the same sites, by the same author. The present study differed in that it added analyses, based on data collected in July 2011, on various canopy and soil metrics.  My main concern upon reading this manuscript is that the various soil nitrogen metrics did not show any correlation with species biodiversity as would be expected if nitrogen fertilization was a major driver behind the biodiversity differences observed between the two plots. As there were only two plots, and, per the author’s description (line 82), there were differences between the plots, it seems feasible that the differences in species diversity may have been due to other plot-specific factors. The author’s assertion that fertilization was lowering biodiversity was therefore not necessarily supported by the findings. I believe that this needs to be explicitly discussed and defended. I was further bothered by the use of soil moisture measurements, taken over a relatively short period of time (a month), being applied to a whole growing season/environment. I would recommend either dropping or defending the latter as well.

There were various minor issues with grammar, defining acronyms, etc. I did not comment on all of these, but some examples, along with specific comments and questions, can be found below:

Line 1: Consider rewriting the title. As written it is hard to follow.

Line 61: I think the way the objectives were written in lines 14 and 15 were a bit more concise and easier to follow. Basically: (1) temporal relations with N and (2) spatial relationships with N plus various "canopy metrics" and "soil resources". Maybe consider stating what those metrics are?

Line 69: Put EEF as an acronym after Fernow Experimental Forest.

Line 74: Citation for this information? Adams et al. 2006?

Line 76: Are the plots being referred to as WS3 and WS4 for consistency with other studies? If so, maybe briefly say so, if not, maybe just refer to them as control and treatment or some other descriptor. It seems unnecessary to make the reader remember which is which.

Line 80: This is not very clear. Was the 7.1 kg ha-1 N apart of the 33.6 kg ha-1 of fertilizer? Were they separate? What else was in the fertilizer? If there was phosphorus/potassium/micronutrients it should be discussed.

Line 82: Does WS3 not support even-aged stands >100 years? This sentence suggests differences between sites.

Line 93: More information here, please (“previously described method”…)

Line 95: More information here, please (“visual method”…)

Line 99: In brief, please explain how these data were collected. How often was soil moisture data collected? With what and how?

At and around line 109: I am not following. "When penetrating canopy openings to the sky"? Please consider rewriting the sentence for clarity. What specifically constituted an "out-of-range" value? Was there a cut-off value that was presumed an inaccurate measurement? If so, why? Please elaborate.

Line 116: Company or other information for CANOCO?

Line 118: define acronyms (Detrended Correspondence Analysis)

Line 129: Please add a citation for this section.

Line 132: Why not use a mixed model?

Line 196. Italicize species name throughout.

Line 204: Consider a few sentences discussing Hills N2 (a brief description of what it is here).

Line 214: Can you give more of an explanation on what % versus time means and how to interpret?

Line 218: Soil moisture can vary considerably through time; to a lesser extent so can CAI and some of the other metrics listed here. Can you really defend that these measurements are indicative of the growing season, or environment as a whole? In the case of soil moisture, I would suggest dropping if all you have is a month of data.

Line 227: Why not just spell out gap function and rugosity in equation?

Lines 217 to 228 would be more appropriate in the methods section.

Line 248: Lines 243 to 248 should be in the methods section.

Line 296: Did you mean “influencing”?

Author Response

Author responses to reviewer comments and suggestions

Although I am pleased with the generally positive comments from both revisions, I am far more appreciative of the numerous suggestions for improvements regarding my manuscript.  Indeed, I feel that making virtually of these changes has improved its clarity and meaning.  Below are my specific responses given in red font.

Reviewer 2

Comments and Suggestions for Authors

The present study examined spatial and temporal differences in herb biodiversity and cover, over 26 years, in two watersheds (or plots), one of which received fertilization multiple times a year, the other did not. The study further examined the relationships between various soil N metrics, soil moisture, various canopy metrics, and herb biodiversity and cover. From this study, the author was able to draw some interesting and meaningful conclusions; including that the addition of nitrogen lowers herb biodiversity, and herb cover was largely driven by nitrogen limitations in the control plot and light limitations in the plot receiving fertilization (as nitrogen was no longer limiting). The use of DCA in the analysis was appropriate.

Again, these positive comments are appreciated.

The effects of additional N on herb layer biodiversity and cover was previously reported in the manuscript “Twentyfiveyear response of the herbaceous layer of a temperate hardwood forest to elevated nitrogen deposition”, on the same sites, by the same author. The present study differed in that it added analyses, based on data collected in July 2011, on various canopy and soil metrics.  My main concern upon reading this manuscript is that the various soil nitrogen metrics did not show any correlation with species biodiversity as would be expected if nitrogen fertilization was a major driver behind the biodiversity differences observed between the two plots. As there were only two plots, and, per the author’s description (line 82), there were differences between the plots, it seems feasible that the differences in species diversity may have been due to other plot-specific factors. The author’s assertion that fertilization was lowering biodiversity was therefore not necessarily supported by the findings. I believe that this needs to be explicitly discussed and defended. I was further bothered by the use of soil moisture measurements, taken over a relatively short period of time (a month), being applied to a whole growing season/environment. I would recommend either dropping or defending the latter as well.

To clarify what is described in Methods, there were actually seven plots in each of two 35-40 ha watersheds.

There were various minor issues with grammar, defining acronyms, etc. I did not comment on all of these, but some examples, along with specific comments and questions, can be found below:

Line 1: Consider rewriting the title. As written it is hard to follow. Done

Line 61: I think the way the objectives were written in lines 14 and 15 were a bit more concise and easier to follow. Basically: (1) temporal relations with N and (2) spatial relationships with N plus various "canopy metrics" and "soil resources". Maybe consider stating what those metrics are? Changed

Line 69: Put EEF as an acronym after Fernow Experimental Forest. Done

Line 74: Citation for this information? Adams et al. 2006? Added

Line 76: Are the plots being referred to as WS3 and WS4 for consistency with other studies? If so, maybe briefly say so, if not, maybe just refer to them as control and treatment or some other descriptor. It seems unnecessary to make the reader remember which is which. Yes, used for consistency.

Line 80: This is not very clear. Was the 7.1 kg ha-1 N apart of the 33.6 kg ha-1 of fertilizer? Were they separate? What else was in the fertilizer? If there was phosphorus/potassium/micronutrients it should be discussed. As described in Methods, fertilizer was ammonium sulfate.

Line 82: Does WS3 not support even-aged stands >100 years? This sentence suggests differences between sites. Correct

Line 93: More information here, please (“previously described method”…) Done

Line 95: More information here, please (“visual method”…) Done

Line 99: In brief, please explain how these data were collected. How often was soil moisture data collected? With what and how?

At and around line 109: I am not following. "When penetrating canopy openings to the sky"? Please consider rewriting the sentence for clarity. What specifically constituted an "out-of-range" value? Was there a cut-off value that was presumed an inaccurate measurement? If so, why? Please elaborate. Wording as per methodology.

Line 116: Company or other information for CANOCO? Done

Line 118: define acronyms (Detrended Correspondence Analysis) Done

Line 129: Please add a citation for this section. Done

Line 132: Why not use a mixed model? This method has been used successfully in other aspects of this project.

Line 196. Italicize species name throughout. Done

Line 204: Consider a few sentences discussing Hills N2 (a brief description of what it is here). This is a pretty standard metric of diversity.

Line 214: Can you give more of an explanation on what % versus time means and how to interpret? Done

Line 218: Soil moisture can vary considerably through time; to a lesser extent so can CAI and some of the other metrics listed here. Can you really defend that these measurements are indicative of the growing season, or environment as a whole? In the case of soil moisture, I would suggest dropping if all you have is a month of data. Agreed.  However, what we were addressing was spatial pattern.

Line 227: Why not just spell out gap function and rugosity in equation? Efficiency of space.

Lines 217 to 228 would be more appropriate in the methods section. Agreed, moved.

Line 248: Lines 243 to 248 should be in the methods section. Agreed, moved.

Line 296: Did you mean “influencing”? Yes